# Adherence to COVID-19 Nutrition Guidelines Is Associated with Better Nutritional Management Behaviors of Hospitalized COVID-19 Patients

**DOI:** 10.3390/nu13061918

**Published:** 2021-06-03

**Authors:** Amelia Faradina, Sung-Hui Tseng, Dang Khanh Ngan Ho, Esti Nurwanti, Hamam Hadi, Sintha Dewi Purnamasari, Imaning Yulia Rochmah, Jung-Su Chang

**Affiliations:** 1School of Nutrition and Health Sciences, College of Nutrition, Taipei Medical University, Taipei 110, Taiwan; amelia.faradinaa@gmail.com (A.F.); nganhdk91@gmail.com (D.K.N.H.); 2Department of Physical Medicine and Rehabilitation, Taipei Medical University Hospital, Taipei 110, Taiwan; d301091012@tmu.edu.tw; 3Department of Physical Medicine and Rehabilitation, School of Medicine, College of Medicine, Taipei Medical University, Taipei 110, Taiwan; 4Department of Nutrition, Faculty of Health Sciences, University of Pembangunan Nasional Veteran, Jakarta 12450, Indonesia; estinurwanti@upnvj.ac.id; 5Department of Nutrition, Faculty of Health Sciences, Alma Ata University, Yogyakarta 55183, Indonesia; hhadi@almaata.ac.id (H.H.); sinthadewips@almaata.ac.id (S.D.P.); 6Department of Public Health, Faculty of Health Sciences, Alma Ata University, Yogyakarta 55183, Indonesia; 7Alma Ata Center for Healthy Life and Foods (ACHEAF), Alma Ata University, Yogyakarta 55183, Indonesia; 8Nutrition Department, Hermina Tangkubanprahu Hospital, Malang 651194, Indonesia; rochmah.imaning@gmail.com; 9Graduate Institute of Metabolism and Obesity Sciences, College of Nutrition, Taipei Medical University, Taipei 110, Taiwan; 10Nutrition Research Center, Taipei Medical University Hospital, Taipei 110, Taiwan; 11Chinese Taipei Society for the Study of Obesity (CTSSO), Taipei 110, Taiwan

**Keywords:** COVID-19, nutrition care, guidelines adherence, length of stay, mortality, Indonesia

## Abstract

Good nutritional support is crucial for the immune system to fight against coronavirus disease 2019 (COVID-19). However, in the context of a pandemic with a highly transmissible coronavirus, implementation of nutrition practice may be difficult. A multicenter electronic survey involving 62 dieticians was conducted, in order to understand barriers associated with dieticians’ adherence to nutrition guidelines for hospitalized COVID-19 patients in Indonesia. 69% of dieticians felt under stress when performing nutrition care, and 90% took supplements to boost their own immunity against the coronavirus. The concerns related to clinical practice included a lack of clear guidelines (74%), a lack of access to medical records (55%), inadequate experience or knowledge (48%), and a lack of self-efficacy/confidence (29%) in performing nutritional care. Half (52%) of the dieticians had performed nutrition education/counseling, 47% had monitored a patient’s body weight, and 76% had monitored a patient’s dietary intake. An adjusted linear regression showed that guideline adherence independently predicted the dieticians’ nutrition care behaviors of nutrition counselling (ß: 0.24 (0.002, 0.08); *p* = 0.04), and monitoring of body weight (ß: 0.43 (0.04, 0.11); *p* = 0.001) and dietary intake (ß: 0.47(0.03, 0.10); *p* = 0.001) of COVID-19 patients. Overall, adherence to COVID-19 nutrition guidelines is associated with better nutritional management behaviors in hospitalized COVID-19 patients.

## 1. Introduction

Indonesia is among the 20 countries currently most severely affected by coronavirus disease 2019 (COVID-19) worldwide, with the fifth highest observed case–fatality ratio (3.0% per 100 confirmed cases) [1]. The clinical characteristics of COVID-19 are diverse, and symptoms range from asymptomatic, mild with nonspecific symptoms (e.g., fever, cough, sore throat, and headaches), moderate/severe pneumonia with acute respiratory distress syndrome (ARDS) demanding mechanical ventilation, and multi-organ failure to death [2]. Currently, remdesivir is the only antiviral drug approved by the U.S. Food and Drug Administration (FDA) for treating COVID-19. Since there are limited effective antiviral drugs, supportive care with good nutritional support is crucial for the immune system to fight against coronavirus infection in hospitalized COVID-19 patients [2,3,4,5].

Dieticians are an integral part of healthcare systems, and are responsible for assessing the nutritional needs of hospitalized COVID-19 patients. However, one of the practical challenges of nutritional management with COVID-19 is the lack of clear guidelines, as the emerging coronavirus and its impacts on health are constantly evolving [2,3,4,5]. Although nutritional management of COVID-19 disease is, in principle, similar to that of hospitalized patients or patients in intensive care units (ICUs) [6], implementation of nutrition guidelines into clinical practice is a great challenge in the context of this pandemic with the highly transmissible coronavirus [7]. For example, dieticians might not be allowed to meet patients or perform nutritional assessments due to the risk of contracting or transmitting COVID-19. Frequently, some instruments for evaluating nutritional status are not readily available in most settings dedicated to COVID-19 patients. Indeed, dieticians should rely on rapid/alternative measures [8]. With the ongoing pandemic, health workers are burned out and are suffering from psychological symptoms (e.g., depression, anxiety, and insomnia), and these may also affect their motivation to implement nutrition guidelines [9,10]. Currently, little is known about the challenges and barriers that affect dieticians’ implementation of COVID-19 nutritional guidelines. The broad aim of this study was to investigate barriers to dieticians’ adherence to nutritional guidelines in hospitalized COVID-19 patients in Indonesia. Specific aims were: (1) to understand the practical challenges and concerns associated with nutritional care, and (2) to understand barriers (guideline knowledge, attitudes, and environmental factors) associated with nutritional management behaviors of dieticians (as indicated by monitoring a patient’s body weight (BW) and dietary intake, as well as performing nutrition counseling/education).

## 2. Materials and Methods

### 2.1. Study Participants

This study was a multicenter electronic survey designed to understand the barriers associated with dieticians’ adherence to clinical practice of nutrition care for hospitalized COVID-19 patients in Indonesia. This study was conducted during November 2020–January 2021. The link of the questionnaire (as a Google Form) was sent to social media groups of Indonesian dietetic association networks, where an estimated number of 210 of the group members were working as dieticians in a hospital. In total, 62 dieticians from 44 hospitals completed the online questionnaire, giving a response rate of 29.5%. Out of the 44 participating hospitals, 39 (88.6%) were located in Jakarta and Java Island, and had higher COVID-19 cases compared to other regions in Indonesia. In addition, 20 participating hospitals (45%) were hospitals designated for COVID-19 by the Indonesia Ministry of Health. The study was conducted anonymously, and no personal data were collected (e.g., name or contact address). Participants were informed of the purpose of the online survey, and their consent to participate in the study was assumed if they completed the online survey. Each participant was allowed to complete the online survey only once. Participants were included if they were of Indonesian nationality, were employed as a dietician in a hospital, had performed nutritional care for hospitalized COVID-19 patients, and completed the online surveys. Exclusion criteria were a non-Indonesian nationality, dieticians who never performed nutritional care for hospitalized COVID-19 patients, and those who did not complete the online survey questionnaires. The study was approved by the Research Ethic Committee of Alma Alta University, Indonesia (KE/AA/XI/10323/EC/2020).

### 2.2. Survey Questionnaire: Barriers to Dietician Adherence to Nutrition Care for Hospitalized COVID-19 Patients

The questionnaire was developed based on the framework of “barriers to physician adherence to practice guidelines in relation to behavior change”, which was proposed by Cabana et al. and published in the Journal of the American Medical Association (JAMA) in 1999 [11]. The questionnaire consisted of four domains: knowledge (12 questions), attitudes (six questions), environmental factors (seven questions), and behaviors (three questions) (Appendix A). Depending on a participant’s answers, each question was awarded 1 or 0 points, with a maximum of 28 points in total (Appendix A). For example, 1 point was awarded to a participant if they know “ESPEN guidelines on clinical nutrition in the intensive care unit” [6] or if they had “monitored the body weight of hospitalized COVID-19 patients”. A higher total score of guideline knowledge, attitudes, environment, and behavior indicates better dietician adherence to clinical nutrition practices for hospitalized COVID-19 patients.

The “guideline knowledge section” (12 questions in total) included awareness of the guidelines (four questions) and familiarity with clinical nutrition practice of the guidelines (eight questions). The four guidelines were published between February 2019 and July 2020, and included the Coronavirus Disease 2019 (COVID-19) Treatment Guidelines (National Institutes of Health, USA) [2], ESPEN guidelines on clinical nutrition care in the intensive care unit (ICU) [6], ESPEN expert statements and practical guidance for nutritional management of individuals with SARS-CoV-2-infection [3], and Nutrition Therapy in the Patient with COVID-19 Disease Requiring ICU Care (reviewed and approved by the Society of Critical Care Medicine and the American Society for Parenteral and Enteral Nutrition) [5]. Familiarity with clinical practice associated with the guidelines included questions such as “is it important to conduct nutritional screening and nutritional assessment for hospitalized COVID-19 patients?” and “must the nutritional assessment and early nutritional care management of COVID-19 patients be integrated into the overall therapeutic strategy?” Respondents answered with “agree” or “disagree”.

The “attitude section” (six question in total) consisted of two parts: self-efficacy or confidence (three questions) and motivation (three questions) in performing nutritional care for hospitalized COVID-19 patients. Examples of the statements included: “lack of self-efficacy or confidence in performing nutrition care for hospitalized COVID-19 patients?”, and “feel stress when performing nutrition care for hospitalized COVID-19 patients?” Respondents answered with “agree” or “disagree”. “Environmental factors” included seven questions including “lack of time, lack of resources, limited budget, limited food supply, lack of access to meet hospitalized COVID-19 patients, lack of access to medical records, and inadequate authority to perform nutritional care for hospitalized COVID-19 patients”. “Dieticians’ behavior” mainly focused on three nutrition care behaviors: (1) “Do you give nutrition education/counseling to hospitalized COVID-19 patients? If yes, how do you give nutrition education/counseling: educational video, educational leaflet, phone call, or text message?”; (2) “Do you monitor COVID-19 patient’s body weight change? If yes, who monitors body weight and how do you do it?”; and (3) “Do you monitor dietary intake of hospitalized COVID-19 patients? If yes, who monitors it and how do you do it?” Total guidelines adherence score (maximum 28 points) was defined as knowledge (12 points), attitudes (six points), environmental factors (seven points), and dieticians’ nutrition practice behaviors (three points). A high total score indicated a better adherence to nutrition guidelines for hospitalized COVID-19 patients. 

### 2.3. Primary Outcome

The primary outcomes were dieticians’ behaviors of nutrition care and self-efficacy or confidence in providing nutrition care for hospitalized COVID-19 patients. The dieticians’ behaviors of nutrition care included: (1) conducting nutrition counseling/education, (2) monitoring patients’ weight changes, and (3) monitoring patients’ dietary intake.

### 2.4. Data Analysis

Statistical analyses were conducted using SPSS 19 (IBM, Armonk, NY, USA). Continuous data are presented as the mean and standard deviation (SD), and categorical data are presented as the number (*n*) and percentage (%). Differences between two groups were analyzed by an unpaired *t*-test. Chi-squared or Fisher’s exact test was employed to compare proportions. An age, gender, years of practice, and type of hospital-adjusted multivariate linear regression model was employed to examine relationships between dependent variables (dieticians’ nutrition practice behaviors) and potential variables related to guideline adherence (total adherence score and its individual components: knowledge, attitude, and environmental factors). *p* < 0.05 was considered statistically significant.

## 3. Results

### 3.1. Participant Characteristics

Table 1 shows baseline characteristics of study participants. In total, 62 Indonesian dieticians participated in the survey; 89% were female and 56% had ≤5 years of clinical experience. Most participants worked in hospitals located in Jakarta (40%), East Java (21%), and Central Java (16%). All participants (100%) had experience in performing nutritional therapy for hospitalized COVID-19 patients, with 48% conducting nutritional therapy for severely and critically ill patients, 40% for patients with mild and moderate illness, and 12% for asymptomatic patients. However, 69% of dieticians felt stress when performing nutritional therapy for hospitalized COVID-19 patients. Ninety percent of participants took supplements or herbal remedies to boost their own immunity against COVID-19, with 63% taking vitamin C, 45% taking vitamin B complex, 30% taking multivitamins and minerals, and 25% consuming ginger (Table 1).

### 3.2. Concerns Related to Nutritional Practices of COVID-19

Table 2 shows concerns related to clinical practices of nutrition care of COVID-19 patients. The most commonly used nutritional screening tools were malnutrition universal screening tools (MUST) (34%) and malnutrition screening tools (MST) (34%), and nutrition assessments were mainly performed by nurses (58%) and dieticians (40%) (Table 2). Seventy-six percent of participants had monitored a patient’s dietary intake; however, only half had monitored a patient’s weight change (47%) or had provided nutrition education or counseling (52%). Ninety-seven percent of participants had recommended supplements for hospitalized COVID-19 patients, of which vitamin C (61%), vitamin B complex (60%), multivitamins/minerals (48%), zinc (40%), and omega 3 fatty acids (27%) were the most frequently recommended supplements. Sixty-eight percent of participants had experience in designing individual diets for hospitalized COVID-19 patients, with 68% modifying the protein content and 63% modifying the total energy. Concerns related to nutritional practices of hospitalized COVID-19 patients included a lack of clear guidelines (74%), a lack of access to meet COVID-19 patients (55%), inadequate experience or knowledge (48%), a lack of self-efficacy or confidence in performing nutrition care (29%), a lack of resources (29%), a limited food supply (29%), and a limited budget (26%) (Table 2).

### 3.3. Barriers to Dieticians’ Adherence to Nutrition Guidelines for COVID-19

Next, we evaluated barriers to dieticians’ adherence to clinical guidelines (Table 3). More than half of the dieticians were aware of “COVID-19 treatment guidelines” (total: 65%; among those with >5 years of experience: 74%; and among those with ≤5 years of experience: 57%) and “ESPEN guideline on clinical nutrition in the intensive care unit” (total: 58%; among those with >5 years of experience: 48%; and among those with ≤5 years of experience: 66%), but to a lesser extent, “nutrition therapy in the patient with COVID-19 disease requiring ICU care” (total: 35%; among those with >5 years of experience: 37%; and among those with ≤5 years of experience: 34%), and “ESPEN expert statements and practical guidance for nutritional management of individuals with SARS-CoV-2 infection” (total: 24%; among those with >5 years of experience: 26%; and among those with ≤5 years of experience: 23%). Most participants were familiar with knowledge of nutrition practice (95–100%) (Table 3). However, 74% of participants thought that there was a lack of clear guidelines for COVID-19, and this rate was slightly higher among junior dieticians (those with ≤5 years of experience: 83%) than senior (those with >5 years of experience: 63%) (*p* = 0.076) (Table 3: Knowledge: Familiarity with clinical practice). Junior dieticians also had lower agreement rates on questions of “I am knowledgeable about the role of nutrition therapy for hospitalized COVID-19 patients” (junior: 43% vs. senior: 81%, *p* = 0.004) and “self-efficacy or confidence in performing nutrition care for hospitalized COVID-19 patients” (junior: 57% vs. senior: 89%, *p* = 0.006), but had a higher rate of “feeling stress when performing nutrition care for hospitalized COVID-19 patients” (junior: 83% vs. senior: 52%, *p* = 0.009). Although 95% of participants agreed that “nutrition counseling is important for hospitalized COVID-19 patients” (Table 3: Knowledge: Familiarity of the guidelines), only half of dieticians (total: 52%) had conducted nutrition education/counseling for hospitalized COVID-19 patients, and this rate was much higher among junior dieticians (junior: 71% vs. senior: 26%; *p* < 0.0001). Only 47% (junior: 54% vs. senior: 37%, *p* = 0.177) had monitored BW changes and 76% (junior: 74% vs. senior: 78%) had monitored dietary intake of hospitalized COVID-19 patients (Table 3: Behavior) 

### 3.4. Factors Predicting Nutrition Care Behaviors of COVID-19 Patients

#### 3.4.1. Self-Efficacy or Confidence in Providing Nutrition Care

Next, we performed a multivariate linear regression analysis to identify factors associated with behaviors of nutrition care for hospitalized COVID-19 patients (Table 4). Age, gender and years of practice-adjusted regression showed that nutrition guideline adherence score (ß: −0.25 (−0.07, −0.01); *p* = 0.03) was negatively correlated with lack of self-efficacy, and, to a lesser extent, disease severity (ß: 0.22 (−0.01, 0.33); *p* = 0.057) (Table 4).

#### 3.4.2. Nutrition Care Behaviors: Nutrition Counseling, and Monitoring of BW and Dietary Intake

A regression analysis adjusted for age, gender and years of practice showed that guideline adherence scores also independently predicted dieticians’ nutrition care behaviors of nutrition counselling (ß: 0.24 (0.002, 0.08); *p* = 0.04), and monitoring of BW (ß: 0.43 (0.04, 0.11); *p* = 0.001) and dietary intake (ß: 0.47(0.03, 0.10); *p* = 0.001) of hospitalized COVID-19 patients (Table 4). Detail analysis of barriers to dieticians’ adherence to nutrition guidelines found that awareness of guidelines was positively correlated with nutrition counselling (ß: 0.70 (0.18, 0.31); *p* < 0.0001), and monitoring patient’s dietary intake (ß: 0.35 (0.03, 0.19); *p* = 0.01). Those dieticians who had better attitude (total score) (ß: 0.03 (0.02, 0.15); *p* = 0.012), self-efficacy or confidence (ß: 0.31 (0.03, 0.26); *p* = 0.013) or motivation (ß: 0.23 (0.02, 0.39); *p* = 0.04) in performing nutrition care were more likely to monitor a patient’s BW (Table 4: adjusted for age, gender, years of practice, and type of hospital).

Next, we investigated the relationship between dieticians’ adherence to nutrition guidelines, length of stay and COVID-19 mortality. Adjusted linear regression analysis showed that guideline awareness was negatively correlated with the length of stay for moderate symptoms (ß: −0.51 (−1.22, −0.14); *p* = 0.017), severe symptoms (ß: −0.31 (−1.48, −0.26); *p* = 0.04) and critical illness (ß: −0.46 (−1.45, −0.16); *p* = 0.029), but not mild symptoms. Guideline familiarity also independently predicted COVID-19 mortality (ß: −40.95 (−63.95, −17.95); *p* = 0.001) (Appendix A).

## 4. Discussion

Our study results indicated that adherence to COVID-19 nutrition guidelines is associated with better nutritional management and, possibly, related to clinical outcome. Studies showed that adherence to nutrition guidelines in critically ill patients is associated with better survival outcomes [12,13]. Currently, Indonesia is not only facing capacity constraints in the health care sector (e.g., man power, funding and facility) but also the unprecedented economic burden of the direct medical cost of COVID-19. It is estimated that median lengths of stay of hospitalized COVID-19 patients were 4~53 days in China and 4~21 days outside of China [14]. In the United States, a single symptomatic COVID-19 infection would cost a direct medical cost of USD 3,045 and one hospitalized case would cost a median of USD 14,366, which only covers costs during the course of the infection and not the follow-up care [15]. The importance of appropriate nutritional assessments and treatments cannot be overstated. The health of COVID-19 patients may rapidly deteriorate after being hospitalized, and patients may develop progressive hypermetabolism 1 week after being intubated in the ICU, which may require 1.6~1.8-times higher energy inputs by the third week post-intubation [16]. Screening and monitoring of a patient’s BW and dietary intake can help doctors and dieticians identify patients at risk of poor outcomes, and also allow planning of individualized nutrition care to support a patient’s immune system in fighting the coronavirus [17]. This is of particular importance for COVID-19, since supportive care is the major treatment method for hospitalized COVID-19 patients, and most severe and critically ill COVID-19 patients are at risk of malnutrition [18,19]. 

Awareness of guidelines also predicts a dietician’s adherence to nutrition guidelines for COVID-19. In the context of a constantly evolving and highly contagious coronavirus, implementation of nutrition guidelines might not be straightforward. Dieticians need to quickly adapt to a wide range of work environments and upgrade their nutrition care programs through training, self-study, or discussing practical problems in real-time through online social networks with fellow dieticians to provide optimal service to COVID-19 patients. Our study found that major concerns related to the nutrition care of COVID-19 patients were a lack of clear guidelines (74%), a lack of self-efficacy (29%), and inadequate experience or knowledge (48%). Dissemination of COVID-19 guidelines with their management algorithm may improve dieticians’ knowledge and promote adherence to guidelines. However, passive dissemination of guidelines might not be effective in the context of the ever-changing COVID-19 pandemic, as the guidelines need to be adapted to local healthcare environments. It is likely that active dissemination or targeted approaches together with supportive networks would improve awareness of, and adherence to, guidelines. For example, Canadian dieticians launched a “COVID-19 response group” on Facebook for dieticians and nutrition students to discuss nutrition care issues, share experiences, and seek advice. Online supportive networks may be particularly important for junior dieticians as our study showed that they had lower self-efficacy/confidence and knowledge than senior dieticians.

Currently, Indonesian hospitals are overwhelmed by COVID-19 and our study found that most Indonesia dieticians, in particularly junior dieticians, are suffering from psychological stress when performing nutritional care for hospitalized COVID-19 patients. Increased psychological stress among junior dieticians is likely due to the combination factors of a higher rate of performing nutritional counseling and a lack of self-efficacy/confidence in performing nutritional care for hospitalized COVID-19 patients. The current study found that psychological stress not only predicted dieticians’ self-efficacy/confidence but also their behaviors of nutrition care of COVID-19. Lu and Dollahite showed that years of nutrition counselling experience significantly predicted self-efficacy scores [20]. Currently, we do not know why Indonesian junior dieticians had a higher rate of performing nutritional counseling for hospitalized COVID-19 patients than senior dieticians, despite the lack of clinical experience. Another interesting finding is that most of dieticians (90%) took supplements as well as recommending supplements (Vitamins C and B complex, multivitamins and zinc) to COVID-19 patients, despite the fact that the COVID-19 Treatment Guidelines stated that there are insufficient data for the panel to recommend the use of vitamins or minerals for the treatment of COVID-19 [2]. Using Google Trends to analyze worldwide concerns with immune-boosting nutrients/herbs during the COVID-19 pandemic, our previous study found that vitamin C, D, E and zinc were the most searched nutrients during the first wave of COVID-19 pandemic [21]. Vitamins and minerals have anti-inflammatory and antioxidant properties, which may support a healthy immune system against coronavirus infection. However, the effects of vitamin and mineral supplementation on COVID-19 remain inconclusive [22,23]. It is very important to prevent or treat nutritional deficiencies. However, supplementation with a supraphysiologic or supratherapeutic amount of micronutrients has not been recommended in the prevention or improvement of clinical outcomes of COVID-19 infection. Therefore, the provision of daily allowances for vitamins and trace elements has been suggested [3,24].

Our study found that environmental factors such as a lack of access to meet COVID-19 patients in person was not a barrier to nutrition care practice. To overcome physical barriers, Indonesian dieticians have employed telemedicine to perform nutrition counseling and monitor patients’ food intake and weight changes. However, feeling stress when independently performing nutrition care predicts the behavior of monitoring a patient’s BW. This suggests that, even when upgrading one’s skills through telehealth channels, dieticians still suffer from psychological stress when dealing with COVID-19. Health organizations need to identify sources of stress and adapt their clinical practice to support nutrition care. Another barrier that predicts the behavior of monitoring a patient’s food intake is the lack of access to medical records. Nutrition care might not be considered a priority in the COVID-19 pandemic, as acknowledged by Thibault and colleagues [7]. Based on their experiences with the COVID-19 pandemic in France, those authors emphasized the need to adapt protocols of nutrition care that are simple and easily applied [7]. Overall, our study results suggest that dieticians need to upgrade their skills in telemedicine and adapt to the local healthcare environment in order to strategize plans for performing individualized nutrition care during the ever-changing COVID-19 pandemic.

The strength of this study includes its novelty, as it is the first to investigate barriers affecting COVID-19 nutrition care, as well as being a multicenter survey with all participants having experience in nutrition care of hospitalized COVID-19 patients. The present study also has several limitations. Firstly, there was a relatively small sample size (*n* = 62) with only one country surveyed (Indonesia) and a low response rate (29.5%). We recognized that a regional study with small sample size may not provide a complete picture of dietetic practice in Indonesia and other countries during the COVID-19 outbreak. The low response rate in our study is due, in part, to the exclusion of dieticians who never performed nutritional care for hospitalized COVID-19 patients in Indonesia. The COVID-19 outbreak itself may also contribute to the low response rate. A recent study in UK also found a limited number of dietitians was able to participate in the online survey due to COVID-19 outbreak, though no response rate was reported [25]. Secondly, information was collected online and not through face-to face interviews. Limitations of online surveys have been noted and intensively discussed [26]. The major strengths of the online survey were its cost effectiveness and the ability to be conducted in a short period of time with no regional restrictions; however, there were concerns about internet accessibility, a lack of control of the sampling or response rate, and ethical issues (e.g., consent, anonymity, and confidentiality) [26]. Nonetheless, it was performed in the context of social distancing during the COVID-19 pandemic, and consent was obtained through participation in the online survey, and all responses were anonymous; the research ethics committee in Indonesia approved the current study. Other limitations include the fact that more confounding factors are needed for the linear regression model when analyzing the relationship between the predictive effect of dieticians’ adherence to nutrition guidelines and the clinical outcomes (survival and length of stay).

## 5. Conclusions

Our study results indicate that adherence to COVID-19 nutrition guidelines is associated with better nutritional management and, possibly, better clinical outcomes. A further validation study is needed in order to provide some definitive guidance on how to implement nutrition guidelines, as well as how the adherence to COVID-19 nutrition guidelines may affect medical cost and economy during the ever-changing COVID-19 pandemic.

## Figures and Tables

**Table 1 nutrients-13-01918-t001:** Characteristics of the study participants (*N* = 62).

Characteristic	Responses
Hospital Characteristic	
**Type of hospital (*n*, %)**	
Government hospital	23 (52%)
Private hospital	21 (48%)
**Region of hospital (*n*, %)**	
Yogyakarta and Central Java	10 (23%)
East Java	11 (25%)
Jakarta	12 (27%)
West Java	5 (11%)
Bali and others	6 (14%)
**Number of hospitalized COVID-19 patients**	14,898.69 ± 23,441.78
**Mortality rate (n, ratio)**	1186 (0.02)
**Average length of stay of COVID-19 patients (day)**	19.58 ± 1.61
Asymptomatic	N/A
Mild Illness	12.58 ± 1.61
Moderate Illness	16.04 ± 1.55
Severe Illness	21.50 ± 2.13
Critical Illness	27.54 ± 2.64
**Dieticians’ characteristics**	
Age (years)	29.27 ± 6.10
Female (*n*, %)	55 (89%)
**Years of practice**	
<1 year	16 (26%)
1~5 years	19 (31%)
5~10 years	14 (23%)
>10 years	13 (21%)
**Have you ever performed nutrition therapy for COVID-19 patients? (yes)**	62 (100%)
**Stages of COVID-19 patients treated? (*n*, %)**	
Asymptomatic	4 (6%)
Mild and moderate illness	20 (32%)
Severe and critical illness	38 (61%)
**Feel stress when performing nutritional therapy for COVID-19 patients?**	43 (69%)
**Take supplements to boost your own immunity against COVID-19?**	56 (90%)
B complex	25 (45%)
Vitamin C	35 (63%)
Multivitamins and minerals	17 (30%)
Ginger	14 (25%)

Continuous variables are presented as the mean ± standard deviation (SD), and categorical data as the number (*n*) (percentage). Mortality rate (case fatality rate) was defined as the number of deaths divided by the number of confirmed cases.

**Table 2 nutrients-13-01918-t002:** Nutrition practice and concerns related to hospitalized COVID-19 patients (*N* = 62).

Nutritional Practice	Responses
**Nutritional screening tools used?**	
Nutrition Risk Screening-2002 (NRS-2002)	7 (11%)
Mini Nutritional Assessment (MNA)	12 (19%)
Malnutrition Universal Screening Tools (MUST)	21 (34%)
Subjective Global Assessment (SGA)	4 (6%)
Malnutrition Screening Tools (MST)	21 (34%)
**Who performs nutritional screening for COVID-19 patients?**	
Dietitian	25 (40%)
Doctor	1 (2%)
Nurse	36 (58%)
**Monitor weight change in COVID-19 patients? (yes: *n*, %)**	29 (47%)
**If yes, who monitors it?**	
Dietitian	21 (34%)
Nurse	6 (20%)
Self-reported by patient	2 (3%)
**Monitor dietary intake of COVID-19 patients? (yes: *n*, %)**	47 (76%)
**If yes, who monitors it?**	
Dietitian	28 (35%)
Nurse	13(27%)
Health care	4 (8%)
Reported by patient	6 (10%)
**Performed nutritional counseling for COVID-19 patients? (yes: *n*, %)**	32 (52%)
**If yes, how do you do it?**	
Educational leaflet	8 (13%)
Phone call	19 (31%)
Text message	10 (16%)
Meet the patient in person	4 (6%)
Video call	1 (2%)
Give education to the family	1 (2%)
**Recommend supplements for COVID-19 patients? (yes: *n*, %)**	60 (97%)
B complex	37 (60%)
Vitamin C	38 (61%)
Multivitamins and minerals	30 (48%)
Zinc	25 (40%)
Omega-3 fatty acids	17 (27%)
**Designed individual diets for hospitalized COVID-19 patients? (yes: *n*, %)**	42 (68%)
Modify total energy	39 (63%)
Modify carbohydrate content	15 (24%)
Modify protein content	42 (68%)
Modify lipid content	10 (16%)
Modify fruits and vegetables	20 (32%)
Give supplements	13 (21%)
No differences	7 (11%)
**Confidence in performing nutritional support for COVID-19 patients with poly-comorbidities (5: very confident; 3: slightly confident; 1: not confident)**	3.37 ± 0.96
**Concerns related to nutrition care for COVID-19 patients**	
Lack of clear guidelines	46 (74%)
Lack of self-efficacy or confidence in performing nutritional care	18 (29%)
Inadequate experience or knowledge	30 (48%)
Limited budget	16 (26%)
Lack of time	7 (11%)
Lack of resources	18 (29%)
Limited food supply	18 (29%)
Lack of access to meet COVID-19 patients	34 (55%)
Lack of access to medical records	9 (15%)

Continuous variables are presented as the mean ± standard deviation (SD). Categorical variables are presented as number (*n*) (percentage).

**Table 3 nutrients-13-01918-t003:** Barriers to dietician adherence to nutritional guidelines in relation to nutritional practice behaviors of hospitalized COVID-19 patients.

Barriers	Total	Years of Practice		*p* Value *
	(*N* = 62)	≤5 Years (*N* = 35)	>5 Years (*N* = 27)	
**Knowledge**				
Awareness of guidelines				
ESPEN guidelines on clinical nutrition in the intensive care unit [6]	36 (58%)	23 (66%)	13 (48%)	0.165
ESPEN expert statements and practical guidance for nutritional management of individuals with SARS-CoV-2-infection (Europe) [3]	15 (24%)	8 (23%)	7 (26%)	0.780
Nutrition Therapy in Patients with COVID-19 Disease Requiring ICU Care (reviewed and approved by the Society of Critical Care Medicine and the American Society for Parenteral and Enteral Nutrition) [5]	22 (35%)	12 (34%)	10 (37%)	0.822
Coronavirus Disease 2019 (COVID-19) Treatment Guidelines (National Institutes of Health, USA) [2]	40 (65%)	20 (57%)	20 (74%)	0.167
Familiarity with the guidelines				
The nutritional assessment and early nutritional care management of COVID-19 patients must be integrated into the overall therapeutic strategy	62 (100%)	35 (100%)	27 (100%)	NA
It is important to conduct nutritional screening and nutritional assessment of hospitalized Covid-19 patients	61 (98%)	34 (97%)	27 (100%)	0.376
It is important to monitor the body weight change in hospitalized COVID-19 patients	48 (77%)	27 (77%)	21 (78%)	0.953
It is important to monitor the dietary intake of hospitalized COVID-19 patients	62 (100%)	35 (100%)	27 (100%)	NA
Nutrition therapy plays an important role in the outcomes of COVID-19 treatment	62 (100%)	35 (100%)	27 (100%)	NA
Nutrition supplementation is useful for treating COVID-19 patients	60 (97%)	33 (94%)	27 (100%)	0.207
Nutrition counseling is important for COVID-19 patients	59 (95%)	32 (91%)	27 (100%)	0.119
Lack of clear guidelines	46 (74%)	29 (83%)	17 (63%)	0.076
**Attitudes**				
Self-efficacy/confidence in performing nutritional care				
I am knowledgeable about the role of nutrition therapy for COVID-19 patients	37 (60%)	15 (43%)	22 (81%)	0.004
Self-efficacy or confidence in performing nutrition care for hospitalized COVID-19 patients	44 (71%)	20 (45%)	24 (89%)	0.006
I have adequate knowledge to design meals for hospitalized COVID-19 patients	32 (52%)	16 (50%)	16 (50%)	0.290
Motivation in performing nutritional care				
I regularly make decisions regarding nutrition therapy as part of the management of COVID-19 patients	50 (81%)	28 (80%)	22 (81%)	0.884
I have an obligation to improve the health of COVID-19 patients by discussing nutrition with them	59 (95%)	33 (94%)	26 (96%)	0.715
I feel stress when performing nutrition care for hospitalized COVID-19 patients	43 (69%)	29 (83%)	14 (52%)	0.009
**Environmental factors**				
Lack of time	7 (11%)	4 (11%)	3 (11%)	0.969
Lack of resources	18 (29%)	11 (31%)	7 (26%)	0.636
Limited budget	16 (26%)	8 (23%)	8 (30%)	0.546
Limited food supplies	18 (29%)	13 (37%)	5 (19%)	0.109
Lack of access to meet hospitalized COVID-19 patients	34 (55%)	16 (46%)	18 (67%)	0.100
Lack of access to medical records	9 (15%)	4 (11%)	5 (19%)	0.432
Inadequate authority to perform nutritional care for hospitalized COVID-19 patients	4 (6%)	3 (9%)	1 (4%)	0.439
**Nutritional practice behaviors**				
Perform nutrition education or counseling for hospitalized COVID-19 patients	32 (52%)	25 (71%)	7 (26%)	<0.0001
Monitor body weight of hospitalized COVID-19 patients	29 (47%)	19 (54%)	10 (37%)	0.177
Monitor dietary intake of hospitalized COVID-19 patients	47 (76%)	26 (74%)	21 (78%)	0.502

All variables are expressed as the number (*n*), percentage (%). * The *p* value was analyzed using unpaired Student’s *t*-test for continuous variables or Chi-squared test for categorical variables.

**Table 4 nutrients-13-01918-t004:** Adjusted multivariate regression coefficient (ß) and 95% confidence intervals (CIs) of barriers of nutrition practice behaviors of COVID-19 patients.

Variables	Lack of Self-Efficacy *	*p*-Value	Nutrition Counseling *	*p*-Value	Monitor Body Weight *	*p*-Value	Monitor Dietary Intake *	*p*-Value
Disease severity	0.22 (−0.01, 0.33)	0.057	0.24 (−0.02, 0.41)	0.077	0.05 (−0.17, 0.25)	0.690	0.15 (−0.09, 0.28)	0.286
Type of hospital	−0.07 (−0.29, 0.15)	0.527	0.05 (−0.20, 0.30)	0.674	0.03 (−0.29, 0.24)	0.844	0.11 (−0.32, 0.14)	0.435
**Total adherence score**	−0.25 (−0.07, −0.01)	0.030	0.24 (0.01, 0.08)	0.040	0.43 (0.04, 0.11)	0.001	0.47 (0.03, 0.10)	0.001
**Knowledge (total score)**	−0.15 (−0.12, 0.03)	0.209	0.19 (−0.03, 0.15)	0.157	0.13 (−0.04, 0.13)	0.287	0.05 (−0.06, 0.09)	0.708
Guideline awareness	−0.01 (−0.08, 0.08)	0.969	0.70 (0.18, 0.31)	<0.0001	0.15 (−0.04, 0.15)	0.273	0.35 (0.03, 0.19)	0.010
Guideline Familiarity	−0.05 (−0.22, 0.14)	0.666	0.11 (0.13, 0.33)	0.402	0.01 (−0.19, 0.18)	0.936	0.03 (−0.22, 0.17)	0.173
**Attitude (total score)**	NA		0.07 (−0.08, 0.14)	0.584	0.15 (−0.30, 0.13)	0.210	0.03 (0.02, 0.15)	0.012
Self-efficacy or confidence	NA		0.05 (−0.45, 0.03)	0.660	0.08 (−0.10, 0.19)	0.643	0.31 (0.03, 0.26)	0.013
Motivation	−0.18 (−0.04, 0.35)	0.112	0.03 (−0.26, 0.20)	0.800	0.07 (−0.17, 0.30)	0.568	0.23 (0.02, 0.39)	0.040
Feel stress	0.23 (−0.48, 0.31)	0.080	−0.37 (−0.67, −0.12)	0.006	−0.24 (−0.57, 0.04)	0.091	−0.21 (−0.46, 0.08)	0.172
**Environmental factor** **(total score)**	−0.15 (−0.15, 0.03)	0.186	0.08 (−0.15, 0.08)	0.535	0.15 (−0.04, 0.18)	0.217	0.12 (−0.06, 0.14)	0.384

Total adherence score (maximum 28 points) was defined as knowledge (12 questions), attitudes (six questions), environmental factors (seven questions), and behaviors (three questions). * Results were adjusted for age, gender, years of practice, and type of hospital.

## Data Availability

The data are not publicly available due to participant confidentiality.

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
