# Peer review of "Adherence to COVID-19 Nutrition Guidelines Is Associated with Better Nutritional Management Behaviors of Hospitalized COVID-19 Patients"

_nutrients, 2021, doi:10.3390/nu13061918_

Round 1

Reviewer 1 Report

In this original manuscript, the authors conducted a survey to explore adherence to COVID-19 nutrition guidelines and clinical outcomes in COVID-19 patients.

Overall it is a good manuscript with a potential clinical meaning. I have some small comments to do.

Tables contain too much information, I suggest to reorganize them.

Page 2; line 64-66: This is true, I suggest adding also that frequently some instruments to evaluate nutritional status are not readily available in most settings dedicated to COVID patients. Indeed, clinicians should rely on rapid/alternative measures. Add this reference: doi.org/10.1038/s41430-020-00795-0

Page 2; Line 340: I suggest to extend this sentence. It is very important to prevent or treat nutritional deficiencies. However, supplementation with a supraphysiologic or supratherapeutic amount of micronutrients has not been recommended in the prevention or improvement of clinical outcomes of COVID-19 infection. Therefore, it has been suggested the provision of daily allowances for vitamins and trace elements. (doi.org/10.1016/j.clnu.2020.03.022; doi.org/10.1007/s12603-020-1400-x)

Reviewer 2 Report

This is the electronic questionnaire survey for dieticians to investigate the nutrition practices and barriers during COVID-19 era in Indonesia. Unfortunately, there were various and strong confounding factors in the analysis for the length of hospital stay and mortality for their patients, therefore, they should not say the relationship to patients' outcomes. Then, this article merely investigated how much the 62 dieticians knew the previous guidelines and what they did in clinical nutrition practice, which could not be validated to the other countries. 

Reviewer 3 Report

“Adherence to Covid-19 nutrition guidelines is associated with better nutritional management behaviors and clinical outcome of hospitalized Covid-19 patients” by

Amelia Faradina et al.

This study is of interest to clinicians working with COVID-19 patients, dieticians, nurses and doctors wording in ICUs. The study is well presented and I have only minor comments.

implementation of nutrition guide- 62 lines into clinical practice is a great challenges in the context of this pandemic with the 63

  • Change challenges to challenge

In total, 62 dieticians from 44 hospi- 85tals completed the online questionnaire giving a response rate of 29.5%. 39

  • The response rate is very low and this should be compared to the rates of similar studies and added to the limitations of the study

The link of the questionnaire (as Google Forms) was sent to social media groups 83

of Indonesia dietetic association network

  • Most ethical committees do not approve online surveys – Also personal data lows are now very strict and do not allow sending questionnaires through Google Forms or even worse to social media. Could you please explain how do you obtain this permission and provide the Research Ethics Committee’s decision? Although the authors are discussing this issue in the limitations of the study, they should also provide the reasons the Ethics Committee was based upon to approve the online survey.

This is in agreement with 281 the result of a systemic review and meta-analysis which showed that adherence to breast 282 cancer guidelines is associated with better survival outcomes [11].

  • This sentence is irrelevant to the study. Please substitute this sentence with the following sentence. "This is in agreement with 281 results of prospective studies, which showed that adherence to nutrition guidelines in critically ill patients is associated with better survival outcomes". [1-2]
  • 1) [If You Get Good Nutrition, You Will Become Happy; If You Get a Bad One, You Will Become an ICU Philosopher. Briassoulis G, Briassoulis P, Ilia S. Pediatr Crit Care Med. 2019 Jan;20(1):89-90. doi: 10.1097/PCC.0000000000001774.]
  • 2) [Nutrition Is More Than the Sum of Its Parts. Briassoulis G, Briassoulis P, Ilia S. Pediatr Crit Care Med. 2018 Nov;19(11):1087-1089. doi: 10.1097/PCC.0000000000001717.]

Round 2

Reviewer 2 Report

Thank you for the revision. I appreciate your efforts and the study significance, however, the multivariable regression analysis conducted in Results 3.5. and table.5 was not scientific enough, because the authors should include more confounding factors which would be possibly associated the clinical outcomes, especially for survival and length of stay. This analysis could not adjusted them enough. As I understand the authors cannot do it with this study set, I strongly recommend that the relevant statements about the association with clinical outcome should be removed in the abstract, mitigate the discussion about it in the discussion section and add it into the limitation. The result 3.5 should be only for reference.  

Author Response

Thank you for your comments. We have removed table 5 and result 3.5 from text (abstract, methods, results), and modified text accordingly. We put table 5 as supplementary table 2 instead. We have revised

(1) title

“Adherence to Covid-19 nutrition guidelines is associated with better nutritional management behaviors of hospitalized Covid-19 patients”

(page 1, line 2-4)

(2) Abstract

Overall, adherence to Covid-19 nutrition guidelines is associated with better nutritional management behaviors of hospitalized Covid-19 patients.

(page 1, line 29-45)

(3) Discussion

“Our study results indicated that adherence to Covid-19 nutrition guidelines is associated with better nutritional management and, possibly, related to clinical outcome.”

(page 12, line 271-272)

(4) Limitation

“Other limitation include more confounding factors are needed for the linear regression model when analyzing relationship between the predictive effect of dietician’s adherence to nutrition guidelines and the clinical outcomes (survival and length of stay).

(page 13, line 371-374)

(5) Conclusion

“Our study results indicated that adherence to Covid-19 nutrition guidelines is associated with better nutritional management and, possibly, better clinical outcome.”

(page 14, line 376-377)
